# The Lung Microbiota and Lung Cancer: A Growing Relationship

**DOI:** 10.3390/cancers14194813

**Published:** 2022-10-01

**Authors:** Maroun Bou Zerdan, Joseph Kassab, Paul Meouchy, Elio Haroun, Rami Nehme, Morgan Bou Zerdan, Gracia Fahed, Michael Petrosino, Dibyendu Dutta, Stephen Graziano

**Affiliations:** 1Department of Internal Medicine, SUNY Upstate Medical University, Syracuse, NY 13210, USA; 2Department of Hematology and Oncology, Cleveland Clinic Florida, Weston, FL 33326, USA; 3Faculty of Medicine, Saint-Joseph University, Beirut 11072180, Lebanon; 4Department of Internal Medicine, Naef K. Basile Cancer Institute, American University of Beirut Medical Center, Beirut 11072020, Lebanon; 5Department of Medicine, Division of Hematology and Oncology, SUNY Upstate Medical University, Syracuse, NY 13210, USA; 6Department of Medicine, University of Pavia, 27100 Pavia, Italy; 7Faculty of Medicine, American University of Beirut, Beirut 11072020, Lebanon

**Keywords:** lung cancer, microbiota, intestinal barrier, dysbiosis

## Abstract

**Simple Summary:**

In the past few years, the microbiota has emerged as a major player in cancer management. The efficacy of chemotherapy or immunotherapy may be influenced by the concomitant use of antibiotics before, during, or shortly after treatment with immune checkpoint inhibitors. Despite this, the mechanism linking the microbiota, host immunity, and malignancies are not clear, and the role of microbiota manipulation and analyses in cancer management is underway. In this manuscript, we discuss the role of the microbiota in the initiation, progression, and treatment outcomes of lung cancer.

**Abstract:**

The lung is home to a dynamic microbial population crucial to modulating immune balance. Interest in the role of the lung microbiota in disease pathogenesis and treatment has exponentially increased. In lung cancer, early studies suggested an important role of dysbiosis in tumor initiation and progression. These results have helped accelerate research into the lung microbiota as a potential diagnostic marker and therapeutic target. Microbiota signatures could represent diagnostic biomarkers of early-stage disease. Lung microbiota research is in its infancy with a limited number of studies and only single-center studies with a significant methodological variation. Large, multicenter longitudinal studies are needed to establish the clinical potential of this exciting field.

## 1. Introduction

The microbiota is a collection of bacteria, fungi, and viruses that reside in or on the human body [1]. It is part of the innate immune system and modulates several mechanisms of host defense [2]. Dysbiosis refers to an alteration in the composition, diversity, or metabolites of the microbiota, which predisposes the body to a variety of diseases [3]. In the past decade, the bulk of microbiota research was confined to the gut and the skin, while the lungs were previously thought to be sterile [4]. In reality, the lungs harbor a community of bacteria (2.2 × 10^3^ bacterial genomes per cm^2^) that largely consists of Bacteroidetes, Firmicutes, Proteobacteria, and Actinobacteria [5,6]. The constitution of this microbiota is determined by the (a) immigration, (b) elimination, and (c) replication rates of the microorganisms [4]. In a healthy lung, microbial immigration and elimination are the predominant regulators of the microbiota [4]. In a diseased lung, the regional growth conditions that affect replication rates are the main processes governing microbiota constitution [4]. Figure 1 shows examples of those dynamics affecting microbiota constitution in a healthy and diseased lung. Host–microbiota interactions have an important role in modulating both innate and adaptive immune responses [7]. While the mechanisms behind the microbiota’s role in immune function have been substantially studied in the gastrointestinal tract, lung microbiota can have similar immune-modulating effects because both the gut and lung environments have similar properties (i.e., mucosal barrier and continuous exposure to the outside) [3].

The interaction between the host microbiota and the innate immune system relies on pattern recognition receptors (PRRs) that attach to microbial antigens. This interaction is mutually beneficial because (a) the microbiota provides a baseline stimulation of the innate immune system while regulating the release of cytokines, and (b) the innate immune system compartmentalizes the commensal bacteria while preventing the overgrowth of pathogenic strains [8]. The cytokines released by the innate immune system are also important for the differentiation of CD4 T cells into pro-inflammatory or regulatory T cells, which further implies that the microbiota can modulate the adaptive immune response [7]. A cohort study by Segal et al. supported this concept by showing that a higher load of supraglottic bacteria in the broncho-alveolar lavage (BAL) was associated with increased inflammatory cytokines and Th17 lymphocytes [9].

## 2. Lung Microbiota and Non-Cancerous Diseases

Dysbiosis in the lungs results from an abnormal alteration in migration, elimination, and/or regional growth factors affecting the microbiota [4]. For instance, gastroesophageal reflux disease increases microbial migration, while impaired mucociliary clearance (e.g., from smoking) decreases microbial elimination [10]. More importantly, dysbiosis from changes in regional growth conditions alters the commensal microorganism population [7]. This change can be reversed if the noxious stimulus is removed, or it can be irreversible if enough inflammatory damage to epithelial cells has occurred [1,11]. The latter situation leads to a self-perpetuating dysbiosis–inflammatory cycle: dysbiosis leads to airway inflammation, which further alters the environmental growth conditions and contributes to dysbiosis [7]. In addition, airway inflammation increases mucus production and permeability, which further worsens the bacterial burden through (a) increasing the supply of nutrients to the lungs and (b) creating areas of increased temperature and hypoxia [4,7]. This promotes the expansion of pathogenic bacterial strains (e.g., Pseudomonas) that are adapted to these conditions [4]. The microbiota has been implicated in the pathogenesis of chronic lung diseases such as chronic obstructive lung disease (COPD), asthma, cystic fibrosis, and idiopathic pulmonary fibrosis (IPF) [12]. All of these diseases are characterized by periods of acute exacerbations that can be induced by dysbiosis [12]. For example, Dicker et al. found that reduced microbial diversity along with Proteobacteria dominance (predominantly *Haemophilus*) was associated with worse COPD phenotypes and more frequent exacerbations [13]. Dysbiosis has been implicated in the development of emphysema. In a study, Li et al. demonstrated that emphysematous changes in mice accelerated after 20 weeks of exposure to intestinal microbiota of COPD patients [14]. However, there is minimal information in the literature on the role of dysbiosis in emphysema in relation to lung cancer development. In asthma, environmental factors (pollen, allergens, and pollutants) can cause bacterial dysbiosis and thereby promote exacerbation by activating inflammatory pathways, promoting bronchoconstriction, and increasing bronchial hyperresponsiveness [15]. Moreover, cystic fibrosis patients with diverse microbiota containing a low abundance but high prevalence of commensals showed the lowest levels of inflammatory markers (IL-1β, IL-8, TNF-α) and the highest levels of forced expiratory volume (FEV)1% predicted [16]. Similarly, in patients with IPF, lung bacterial burden predicted disease progression, and microbiota diversity and composition correlated with increased alveolar profibrotic cytokines [17].

## 3. Lung Microbiota and Lung Cancer Pathogenesis

The airway microbiota has a unique role in the initiation and progression of lung cancer. Germ-free or antibiotic-treated mice are significantly protected from lung cancer development due to *Kras* mutation and *p53* loss [18]. Meanwhile, commensal bacteria were able to stimulate an inflammatory response (by activating γδ T cells) associated with the development of lung adenocarcinoma [18]. McLean et al. hypothesized that dysbiosis could induce carcinogenesis through three pathways: (a) impaired immune balance, (b) chronic inflammation, and (c) the induction of oncogenic pathways [7]. First, dysbiosis can disrupt the baseline stimulation of the immune system in the lungs. On the one hand, the depletion of microbial diversity impairs the priming of antigen-presenting cells, which hinders their ability to react to tumor antigens [11]. On the other hand, bacterial overgrowth causes the overstimulation of the immune system and uncontrolled proliferation of IL-17–producing CD4 helper T cells, which are key factors in lung tumorigenesis [19]. Second, dysbiosis causes chronic inflammation through DNA-damaging metabolites and genotoxins released by the affected commensal organisms [20]. The inflammatory cells stimulated by dysbiosis can also release reactive oxygen and nitrogen species that lead to angiogenesis and carcinogenesis [21]. Finally, several studies found that some species from the microbiota can directly induce oncogenic pathways (Table 1). Apopa et al. showed that PARP1 is upregulated in non-small cell lung cancer (NSCLC) tissues in the presence of the cyanobacterial toxin microcystin (a toxin produced by cyanobacteria) [22]. Tsay et al. also linked *Streptococcus* and *Veillonella* to the induction of the PI3K and Extracellular signal-regulated protein kinase (ERK) pathways involved in lung cancer [23]. Despite these promising results associating specific microbes with carcinogenesis, it is still difficult to differentiate between the microorganisms that are truly inducing oncogenic pathways and those that are just opportunistic colonizers of the tumor microenvironment [7]. Making this distinction is crucial for investigating the microbiota as a therapeutic target for lung cancer.

### 3.1. Lung Cancer Initiation

The microbiota is involved in the pathogenesis of cancer in various ways. Nonetheless, its implication in the development of lung cancer is still not well unraveled. A common hypothesis is that the lung microbiota could be directly oncogenic. Many explanations include the enhancement of mucosal inflammation or the promotion of immune dysregulation [56] (Figure 2). The key mechanism for the oncogenic potential of the lung microbiota remains the development of an inflammatory niche that favors carcinogenesis associated with multifactorial etiologies, including but not limited to environmental influences and the presence of underlying chronic diseases [21]. For example, COPD patients are at a higher risk of lung cancer. This chronic disease induces a state of microbial dysbiosis that allows for increased colonization and sustainment of an ongoing inflammatory state leading to carcinogenesis and cancer development [57]. Pathophysiology is resumed as follows. Bacterial byproducts and N-formyl peptides are potent chemo-attractive molecules that enhance monocyte and neutrophil recruitment, further increasing lung inflammation, loss of alveolar adhesion, and the destruction of lung parenchyma [11]. IL-6, a pro-inflammatory cytokine, and IL-8, a neutrophil recruitment cytokine, both increased in the setting of infection, are involved in tumorigenesis. They activate the NF-κB-1 pathway and promote cancerous cell proliferation, migration, and invasion [58]. Further notable examples include *Helicobacter pylori*, which is hypothesized to be associated with lung cancer development by prompting the production of pro-inflammatory cytokines such as IL-1, IL-6, and TNFa [59]. *Bacteroides fragilis*, a Gram-negative, anaerobic bacilli, activates STAT3 by Type 17 T-helper cells in mice, which supports lung cancer cell proliferation and angiogenesis [60,61]. Bacterial toxins secreted by the lung microbiota may be associated with tumorigenesis and genotoxicity [21]. *Bacteroides fragilis,* for example, is also responsible for the synthesis of the cytolethal distending toxin and the cytotoxic necrotizing factor-1, both of which are associated with failure of the DNA repair system [42]. The chemicals produced by different microorganisms, such as superoxide dismutase, are also involved in genomic instability [62].

Bacterial products may be implicated in carcinogenesis. Inactivated *Escherichia coli* induces Toll-like receptor 4 (TLR4) activation, which favors the proliferation of NSCLC cells in vivo. Hypothesized mechanisms include the upregulation of p38 MAPK and ERK1/2 signaling [63] and the stimulation of calcineurin expression [60,61]. The dysregulated MAPK pathway over-activates RAF and RAS signaling molecules, leading to uncontrolled cell proliferation and resistance to apoptosis via the ERK protein, a transcription factor that mediates survival gene expression. Further supporting evidence to the hypothesis that dysbiosis in the tumor microenvironment may play a role in the metabolism and metabolic signaling of cancerous cells in the lung includes but is not limited to the following: Yu et al. demonstrated that the lung microbiota in lung cancer patients had an altered metabolic profile. There was a decrease in the amount of KEGG modules of energy metabolism and ATP-binding-cassette type transport with increased amino acid and lipid metabolism. These changes impact the expressions in epithelial respiratory cells, promoting carcinogenesis [64,65]. Another important pathway involved in the initiation of lung cancer includes the PI3K signaling pathway. Tsay et al. exposed human adenocarcinoma cells (A549) to bacterial products directly extracted from lung cancer patients that upregulated the genes involved in the PI3K pathway. Such changes are consistent with alternations observed in lung cancer patients as compared to the unaffected population [23]. PI3K activation is an early, reversible event in lung tumorigenesis. The PI3K/AKT pathway is an intracellular signaling pathway that regulates a variety of cellular functions, including cell survival, metabolism, and growth. Extracellular signaling molecules bind to several receptors located in the cell membrane, leading to receptor activation. This activates intracellular PI3K, which in turn converts PIP2 lipids to PIP3. AKT then binds to PIP3, which can be phosphorylated by PDK1 and mTOR at two distinct sites, leading to the full activation of AKT. pAKT phosphorylates several downstream molecules, including BAD, IKK, CREB, FOXO1, and mTORC1, that regulate numerous cellular processes. The dysfunction of this pathway can lead to the overactivation of the pathway and has been linked to numerous pathological processes, including tumorigenesis [66].

The lung microbiota can shape the immune microenvironment and favor tumor growth. Using an autochthonous genetically-engineered mouse model (GEMM) of lung adenocarcinoma, Jin et al. shed light on the role of the microbiota–immune crosstalk involved in inflammation and cancer genesis [18]. Disrupted local microbiota prompted the proliferation and activation of lung γδ T cells (which represent ‘unconventional’ T cells that represent a relatively small subset of T cells in peripheral blood) using an IL-1β and IL-23-dependent mechanism. This subset of T cells produces IL-17 (not to be confused with T-Helper 17 cells), which promotes neutrophil recruitment and inflammation within the tumor microenvironment [18]. The use of systemic antibiotic treatment to reduce microbiota burden and/or the blocking of the γδ T cells signaling pathways (with blocking of IL17 production) reduced tumor growth within the lung [18,67].

### 3.2. Lung Cancer Progression

The microbiota also plays an undoubtable role in lung cancer metastasis. *Streptococcus* and *Veillonella* can promote the metastasis of lung cancer cells by upregulating the ERK and PI3K pathways (as is also seen in the initiation of lung cancer) in epithelial respiratory cells [23]. Wang et al. demonstrated that, in patients infected by *Mycobacteria tuberculosis*, the ubiquitins found in host cells could activate protein–tyrosine–phosphatase A (PTPA), which favors the migration of human lung adenocarcinoma cells [68]. Additionally, inflammation-related mechanisms can further promote the metastasis of lung cancer. Gram-positive Pneumonia favors NSCLC dissemination via TLR2 activation [69]. As for immunity-related mechanisms, decreased bacterial load is associated with lower amounts of T-reg cells and enhanced helper T-cell and NK cell recruitment, resulting in reduced lung cancer metastases [67]. Human lung microbiota can also enhance angiogenesis via numerous inflammatory factors, further promoting the progression of the disease [70]. The described mechanisms of neovascularization include but are not limited to sprouting angiogenesis and vasculogenesis. The microbiota participates in the abovementioned angiogenesis mechanism via numerous mediators. Vascular endothelial growth factor (VEGF), endothelial cells (EC), inflammatory factors, and cells are the main components. Cui et al. demonstrated that mice treated with fecal microbiota transplantation showed an increased expression of VEGF that increased angiogenesis [71]. Moreover, in the CCSPcre/K-rasG12D mouse lung cancer model exposed to non-typable *Haemophillus influenzae* lysate weekly, the proportion of T-Helper 17 cells in the lung was significantly increased [19]. T-Helper 17 cells secrete numerous cytokines, most importantly IL-17, IL-21, and IL-22. IL-17 can promote angiogenesis by favoring the release of angiogenic mediators, cytokines, and chemokines. *Helicobacter pylori* can also promote the vascularization of lung carcinoma. VacA, an exotoxin produced by this incurvated rod, is found in the lung and can favor the release of IL-8 and IL-6 by lung cells [72,73]. In addition to being a neutrophilic chemotactic factor, IL-8 is a promoting factor for tumor angiogenesis [72,73]. *Human papilloma virus* (HPV) type 16 can also promote angiogenesis in lung cancer cells in vitro. Plausible explanations include the PI3K/Akt signaling pathway and c-Jun [72,73]. Finally, a study by Ushio et al. demonstrated that *Mycoplasma*-infected cells have a higher ability to metastasize in vivo than non-*Mycoplasma*-infected cells. It can be due to the changes in the expression of cell surface proteins which promote metastasis in cancer cells, and changes occurring in host cells influenced by *Mycoplasma* infection. Monoclonal antibodies targeting *Mycoplasma* proteins p37 and Ag 243-5 enhanced tumor cell invasiveness and decreased contact inhibition [74].

## 4. Microbiota as a Therapeutic Target in Lung Cancer

The current treatments of lung cancer are categorized into radiotherapy, chemotherapy, immunotherapy, and surgical resection [75]. However, despite the advances in lung cancer treatment, a substantial number of patients are diagnosed with advanced-stage disease and, unfortunately, suffer from poor prognosis and limited treatment options [76]. Therefore, it is essential to identify early diagnosis strategies and optimize the treatment options for lung cancer. The existing investigations about the clinical applications of microorganisms in lung cancer treatment are still in the early stages. However, deciphering the interplay between lung cancer and the lung/gut microbiota might revolutionize lung cancer treatment. In this section, we will summarize recent advances highlighting the microbiota as a therapeutic target for cancers, with a special focus on lung cancer.

### 4.1. Microbiota and Immunotherapy

Programmed death 1 (PD-1), programmed death-ligand 1 (PD-L1), and cytotoxic T lymphocyte-associated protein 4 (CTLA-4) inhibitors are examples of monoclonal antibodies commonly used in cancer immunotherapy, particularly in advanced NSCLC [77,78,79]. These immune checkpoint inhibitors (ICIs) reverse immunosuppression in the tumor microenvironment by inhibiting the interaction of T cell suppressing receptors with their corresponding ligands on cancer cells.

In addition, ICIs modulate the immune system to enhance the capacity of effector T cells, mainly CD8+ cytotoxic T cells, to identify and recognize tumor-specific antigens, inducing cell death [80]. Recent studies have highlighted the role of microbiota as a leading actor in cancer immunotherapy response. Intestinal dysbiosis, whether due to a disease or due to antibiotic use, may negatively affect the immunotherapeutic response [81,82]. Antibiotics, in particular, can have dueling effects on cancer immunotherapy and treatment outcomes [83]. Since the gut microbiota influence local and systemic antitumor immune response by modulating PRRs, PAMPs, and DAMPs, dysbiosis may have a negative impact on immunotherapy. Dysbiosis can also preferentially select infectious strains of *Clostridioides difficile*, vancomycin-resistant enterococci (VRE), extended-spectrum beta-lactamase (ESBL) organisms, and carbapenem-resistant enterococci (CRE) bacteria and *Candida* species, in the gut. These organisms can severely affect treatment and survival outcomes. Alternately, antibiotics can semi-selectively correct dysbiosis that may have originally contributed to the malignancy and improve mucosal immune response to immunotherapy. The supplementation of gut commensals potentiates immunotherapy and decreases tumor burden. The oral gavage of *Bacteroides fragilis* in mice with melanoma stimulated a specific T cell response that enhanced the efficacy of CTLA-4 blockade [84]. Similarly, oral probiotic supplementation of *Bifidobacterium* re-sensitized tumors to anti-PDL1 therapy in melanoma-carrying mice by stimulating anti-tumor CD8+ cytotoxic T cell activity and inducing dendritic cell maturation [85].

Routy et al. analyzed the microbiota of 249 cancer patients diagnosed with either urothelial carcinoma, advanced NSCLC, or RCC on PD-1 immunotherapy. Antibiotics were given to 69 patients for common indications (dental, urinary, and pulmonary infections). Antibiotics-treated patients had reduced overall survival and shorter cancer recurrence time than those who did not receive antibiotics within the PD-1 treatment period. This suggests that antibiotic consumption can decrease the effectiveness of immunotherapy [86]. Commensals *Akkermansia muciniphila* and *Enterococcus hira* were abundant in the stools of PD-1 responders and were most significantly associated with favorable clinical outcomes in NSCLC [86]. PD-1 blockade was reinstated after fecal transfer from recovered patients into antibiotics-treated mice with melanoma. A similar result was observed after colonizing mice intestines with *Akkermansia muciniphila* and *Enterococcus hira* under several conditions of gut dysbiosis. An enhanced CD4 T cell and IL-12 response is probably responsible for the observed outcomes [86]. Other studies have also shown similar effects of gut microbiota on cancer immunotherapy [87,88]. In another study with advanced NSCLC patients with a good response to anti-PD1, higher gut microbiota diversity was observed compared to the non-responders [89]. The better immunotherapy response was due to an enhanced memory CD8+ T cell and natural killer cell subsets in the peripheral circulation of responders [89]. These studies reveal an important relationship between the efficacy of immunotherapy and dysbiosis of gut microbiota, which, if improved, can decrease the tumor burden and improve overall survival.

### 4.2. Microbiota and Chemotherapy

Similar to immunotherapy, studies have shown that the microbiota can positively or negatively influence the effectiveness of chemotherapeutic agents on various cancers, including lung neoplasms.

Among the members of lung microbiota, *Mycoplasma* preferentially colonizes tumors because of the nutrient-rich tumor microenvironment and its suitability for small, cell wall-lacking bacteria [90,91]. In fact, up to 100% of surgically removed lung cancer tissues have been found to be *Mycoplasma*-infected, indicating a strong association between *Mycoplasma* infection and carcinogenesis [92]. Intratumoral *Mycoplasma* decreases the efficacy of the nucleoside analog gemcitabine, a chemotherapeutic drug used in the treatment of NSCLC. *Mycoplasma* possesses a cytidine deaminase (CDA), which contributes to the degradation of gemcitabine [93] and, thus, decreases its efficacy. Moreover, Gammaproteobacteria, which also preferentially colonize tumors, have CDA such as *Mycoplasma*, and thus induce comparable resistance to certain antimetabolites [94].

The Gammaproteobacteria *E. coli* decreases gemcitabine efficacy in cell cultures, as well as abrogated the anti-tumor response to the antinucleotide in colon cancer mice models as compared with non-infected mice, which showed better response to gemcitabine [94]. Nonetheless, the fluoroquinolone antibiotic ciprofloxacin did restore gemcitabine efficacy in mice infected by CDA-producing bacteria, mainly *Mycoplasma* or *Proteobacteria* [94].

Although some microbial families in the lung may confer resistance to cancer therapy, others in the gut are needed for treatments to be fully effective. For instance, lung cancer mice models treated with an antibiotic cocktail, which destroys host commensal microflora, were found to be less responsive to cisplatin and had a significantly reduced survival rate; whereas mice supplemented with *Lactobacillus* probiotics followed by cisplatin therapy had smaller tumors with an improved survival rate [95]. Gene expression studies suggested that antibiotics impair the function of cisplatin by upregulating the expression of VEGF and downregulating the expression of BAX and CDKN1B. This dysregulation is commonly seen in lung cancer patients leading to angiogenesis and the inhibition of apoptosis, thus promoting tumor development [95]. Moreover, the expression of IFN-γ, granzymes (GZMB), and perforins (PRF1) in the CD8+ T cells in these mice models were reduced after antibiotic treatment, suggesting an immunostimulatory role of the commensal microbiota [95]. In opposition, lung cancer mice models treated with the combination of cisplatin and *Lactobacillus* had lower levels of VEGF, higher levels of apoptosis-promoting BAX, and enhanced the expression of CD8 T cells IFN-γ, GZMB, and PRF1; hence slowing tumor progression [95].

Additional evidence in human studies highlights the importance of healthy gut microbiota for effective cancer therapy. The chemotherapeutic agent cyclophosphamide generated better progression-free survival in advanced lung cancer patients whose gut microbiota contains *Enterococcus hirae*. In fact, cyclophosphamide induces damage to the epithelial barrier, leading to the generation of an effective anti-tumor IFN-γ and IL-17 response [96]. *E. hirae* can form a memory T-helper 1 cell response that enhances cyclophosphamide’s action and increase its efficacy.

## 5. Future Directions

The future of microbiota research in lung cancer is very promising and is progressing in two major directions. First, deciphering the microbiota constitution and how it evolves in the processes of lung cancer initiation and progression would help in developing early diagnostic markers. Second, the exploration and exploitation of the host–microbiota interactions may serve as potential targets for lung cancer treatment [24]. Microbiota can be manipulated in three ways to influence lung cancer treatment [97]. Fecal microbiota transplant (FMT) can reverse dysbiosis by restoring normal gut microbiota composition. It also reduces CDI and improves cancer therapy efficacy. Preclinical studies have demonstrated the potential of FMT in improving cancer treatment outcomes in colorectal cancer [98], melanoma [99], NSCLC, and renal cancer [86]. FMT also shows graft-vs-host disease (GvHD) resolution [100,101]. Multiple clinical trials utilizing FMT in patients with malignancies to improve chemotherapy or immune checkpoint inhibitor efficacy in AML, metastatic melanoma, prostate cancer, and renal cell carcinoma are underway.

Antibiotic stewardship can achieve semi-selective modification of intestinal microbiota. Reducing the usage of broad-spectrum antibiotics, especially early in the process of treatment, can be detrimental, as has been observed in both preclinical and clinical settings [87]. Antibiotic stewardship can preserve microbiota diversity and decrease unwanted microbial density and inflammation. For example, prophylactic treatment with the minimally absorbed antibiotic rifaximin is under investigation to assess the reduction of anticancer treatment-associated gastrointestinal toxicity and diarrhea in colon adenocarcinoma (NCT04003181), colorectal cancer (NCT03563586), and stage I-III human epidermal growth factor receptor 2 (HER2)-positive breast cancer (NCT04249622) [97].

The replenishment or enrichment of commensals to create a more balanced microbiota composition can also be achieved using pre- and probiotics. Such biologic modification of the gut microbiota population can improve dysbiosis-related complications in cancer and cancer treatment outcomes. For example, one of the major complications of cancer treatment is the damage to the mucosal barrier that not only diminishes chemo- and immunotherapy effects but also often contributes to bacteremia and candidemia. The probiotic supplementation of bacteria that improves intestinal barrier function can help mitigate such problems. *Akkermansia muciniphila* is a Gram-negative bacterium that fortifies the mucosal layer and maintains gut barrier function. *A. muciniphila* modulates TLR2- expressing cells [102], improve insulin sensitivity [103], and alter the response to anti-PD-1 treatment [86]. Other bacterial species that confer intestinal barrier function improvement by producing short-chain fatty acids are *Faecalibacterium* (butyrate producer) and *Bifidobacterium* (lactic acid and acetic acid producer). Pre- and probiotics that directly or indirectly enrich microbiota with gut barrier function improving bacteria are under investigation to determine their beneficial impact on cancer treatment. For example, *Myrciaria dubia*, a prebiotic that enriches *A. muciniphila* is being investigated for immune checkpoint inhibitions in NSCLC and melanoma patients (NCT05303493). The probiotic *Lactobacillus Bifidobacterium* V9(Kex02) is being explored for improving the efficacy of Carilizumab combined with platinum in NSCLC (NCT05094167). The combined mechanism of FMT and biologic modification of microbiota is in a clinical trial where pooled fecal microbiota with enriched *Faecalibacterium prausnitzii*, *Bifidobacterium longum*, *A. muciniphila*, and *Fusobacterium* spp. is being investigated to improve the effectiveness of PD-1 therapy in advanced lung cancer patients (NCT04924374).

While the prospects of microbiota exploitation in lung cancer treatment are promising, there are concerns regarding the standardization of the procedures [7]. Furthermore, the knowledge about the interactions between various members of the microbiota themselves and between the host cells under normal and stressed conditions such as cigarette smoking [24] is still eluding.

## Figures and Tables

**Figure 1 cancers-14-04813-f001:**
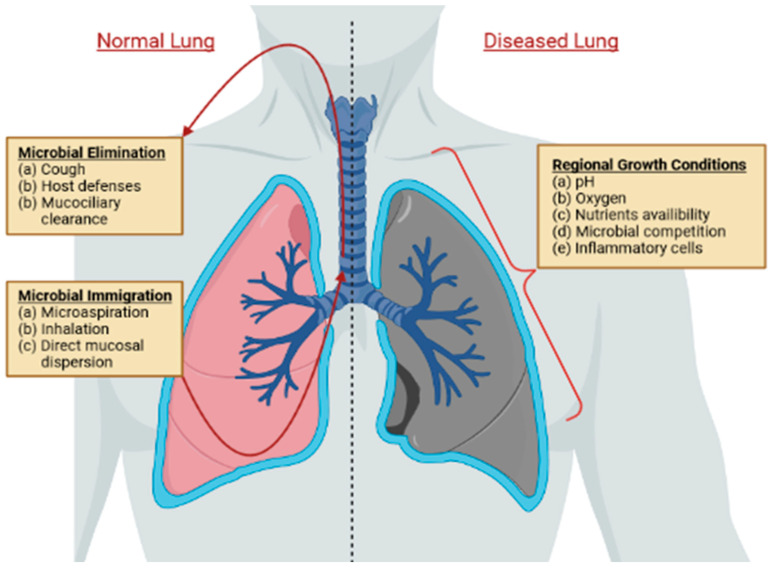
Three factors affecting the dynamics of host microbiota constitution in the lungs: immigration, elimination, and regional growth factors.

**Figure 2 cancers-14-04813-f002:**
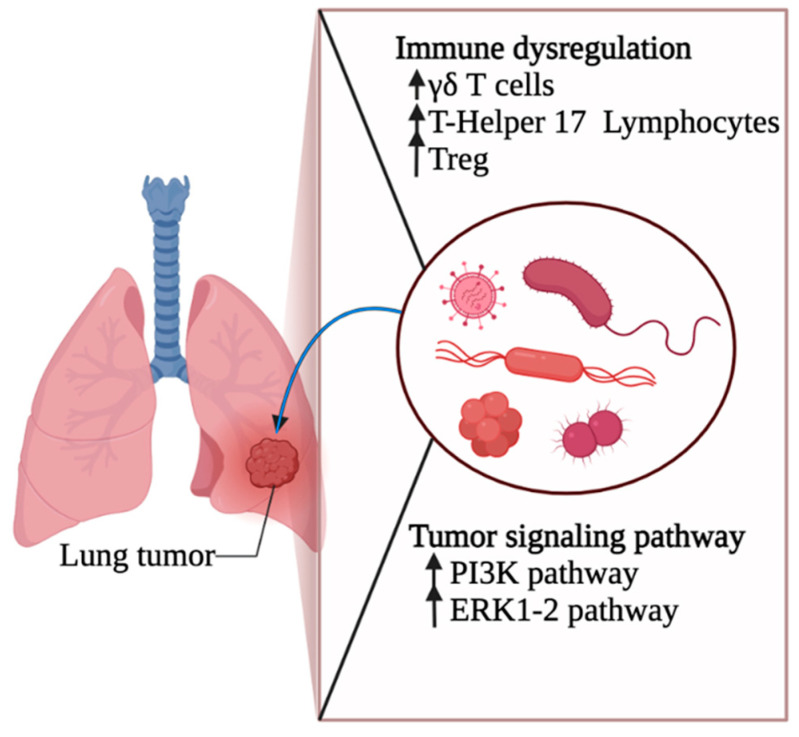
Lung Cancer and Microbiota. Local microbiota plays a critical role in the pathogenesis of lung cancer. Derived metabolites act as signaling molecules, directly amplifying tumor-intrinsic pathways. They can also induce immune dysregulation by favoring inflammation and immunosuppression.

**Table 1 cancers-14-04813-t001:** Lung and Gut Bacterial Species in association to Lung Cancer.

Bacterial Classification	Sampling Method	Association with Lung Cancer	Potential Mechanism of Association with Lung Cancer
*Pseudomonas* [24,25,26,27,28,29]	Lung explants, BAL, Brush specimen, lung tissues	Marked presence in lung adenocarcinoma	Associated with an increase in macrophage population and IFN-γ in BAL sample. Amplification of neutrophil elastase activity. Enhanced/diminished population associated with smoking
*Streptococcus* [24,25,27,30,31,32,33,34]	Lung explants, BAL, Brush specimen, lung tissues	Marked presence in lung cancer, specifically adenocarcinoma, squamous cell carcinoma, and increased risk of hepatic metastasis from NSCLC Increased levels in the gut of small cell lung cancer patients	Enhanced ERK and PI3K pathway. Increased presence of Th17 cells and neutrophils. Enhanced/ diminished population in smokers
*Prevotella* [24,25,30,31,32,33,34]	Lung explants, BAL, Brush specimen	Low presence in the gut of NSCLC patients. Marked presence in the gut of squamous cell carcinoma patients Marked presence in lung cancer and adenocarcinoma	Correlated to an inflammatory phenotype, including an enhanced Th17 lymphocyte and neutrophil response Enhancement of ERK and PI3K pathway
*Fusobacterium* [25,31,32,33,35,36,37]	Lung explants, BAL, Brush specimen	Poor response in lung cancer to anti-PD-1 therapy if *Fusobacterium* is present in the airway. One of the most abundant specific bacterial community members detected in synchronous multiple primary lung cancer sMPLC lesions.	Fap2 protein of *Fusobacterium* inhibits natural killer cell killing by interacting with an inhibitory receptor present on all human NK cells and on various T cells (TIGIT receptor)
*Veillonella* [24,25,27,31,38,39]	Lung explants, BAL, Brush specimen, lung tissues, saliva	Observed in both small cell lung cancer and adenocarcinoma	Correlated to an inflammatory phenotype, including an enhanced Th17 lymphocyte and neutrophil response. Most abundant agent driving dysbiosis and amplification of IL17, PI3K, MAPK, and ERK pathways in the airway transcriptome
*Prophyromonas* [25,26,31,40,41]	Lung explants, BAL, Brush specimen	Higher *P. gingivalis* staining in carcinoma tissues of patients with small cell lung cancer, lung adenocarcinoma, and lung squamous cell carcinoma (35.00%, 26.89%, and 39.00%, respectively) compared to the adjacent lung tissues	Activation of cancer-associated transcription factors by modulating ATP-induced cytosolic, mitochondrial ROS, and antioxidant glutathione response through the inhibition of ATP/ P2X7-induced cell death by *P. gingivalis*.
*Neisseria* [25,42]	Lung explants, BAL, Brush specimen, saliva	Reduced presence in lung cancer	Suppress cell growth
*Haemophilus* [25,26,43]	Lung explants, BAL	Stimulates proliferation of early adenomatous lesions leading to alveolar adenomatous hyperplasia and adenocarcinoma	Upregulation of IL-17C and neutrophil infiltration. Can also promote metastatic progression in combination with cigarette smoke (8)
*Sphingomonas* [26,27]	BAL, Brush specimen, lung tissues	Marked presence in adenocarcinoma	Associated with an increase in macrophage population and prominent IFN-γ population
*Acinetobacter* [25,26,34,44]	Lung explants, BAL	Marked presence in lung cancer and adenocarcinoma	DNA methylation of CpG regions in the promoters of *E-cadherin* gene induced by *A. baumannii* transposase (Tnp) and down-regulation of this gene
*Staphylococcus* [25,28,45]	Lung explants, BAL, Brush specimen, lung tissues	Marked presence in the gut of NSCLC patients responsive to Nivolumab	Lipoteichoic acid induced cellular proliferation and liberation of interleukin (IL)-8.
*Corynebacterium* [25,26,31,46]	Lung explants, BAL, Brush specimen	No significant difference in microbiota composition between ground glass pulmonary nodules and normal tissues except in adenocarcinoma (AD)	-
*Lactobacillus* [24,26,27,31]	BAL, Brush specimen, lung tissues	Low levels in the gut of NSCLC patients (8). High abundance in the gut of squamous cell carcinoma patients	Gut microbiota’s role in regulating the lung’s immune response.
*Actinobacillus* [24,26,31,47]	Lung explants, BAL, saliva	Increase in the gut of lung cancer patients and commonly found in the lungs and sputum of lung cancer patients	*Actinobacillus*’ presence in the airway leads to chronic lung inflammation promoting the initiation and early development of lung cancer.
*Propionibacterium* [26,47]	BAL	Marked presence in the gut of NSCLC patients responsive to Nivolumab Marked presence in squamous cell carcinoma	Gut microbiota’s role in regulating the lung’s immune response.
*Ralstonia* [29,48]	Lung tissues	Marked presence in adenocarcinoma	Plays a role in impairing the tumor microenvironment’s immunity
*Megasphaera* [39,49]	BAL	Marked presence in lung cancer patients	Promotes somatic cell genome instability via high levels of chromosomal aberrations (CAs) and micronuclei (MN) frequency seen in peripheral blood lymphocytes of patients with lung cancer
*Acidovorax* [50,51]	Lung tissues	A marked presence in squamous cell carcinoma patients compared with adenocarcinoma. Specific taxa are more common in smokers with TP53 mutation	Degrades tobacco smoke compounds and thus promoting survival of transformed cells and subsequent tumor development. Malignant transformation of the lung epithelium via DNA damage and mutations in TP53, mediated by microbial toxins or reactive oxygen/nitrogen.
*Capnocytophaga* [52,53,54]	Saliva	A marked increase in lungs of lung cancer patients compared to control	Stimulation of chronic inflammation, thus promoting the development of lung cancer, especially lung squamous cell carcinoma
*Cyanobacteria* [52,55]	Lung tissue	A marked increase in lungs of adenocarcinoma patients	*Cyanobacteria* toxin microcystin is associated with reduced CD36 and increased levels of PARP1

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
