# Peer review of "The Lung Microbiota and Lung Cancer: A Growing Relationship"

_cancers, 2022, doi:10.3390/cancers14194813_

Round 1

Reviewer 1 Report

The manuscript by Zerdan et el. explores the intricate relationship between lung cancer and microbiota with a special focus on the influence of microbiota on the response to immune checkpoint inhibitors. Although the subject is of great interest related to a hot topic in research, some improvements are necessary to be done: 

Major comments:

1. The whole manuscript approaches the subject in a general way. I recommend to authors to insist more on the molecular pathways and mechanisms related to how the microbiota is influencing both lung cancer progression and response to current treatments. Also, as mentioned by you special attention should be given when presenting the data to how the specific bacterial species reached the organ, what is their role and if they are colonization after the development of lung cancer or a promoter and are supporting the lung cancer progression. For example, the fact that mycoplasma was detected in 100% of samples of lung cancer examined in that specific study raises important questions. More details should be given to this subject and the authors should try to be more present in the manuscript. 

2. Change the title of the subchapter 2. from discussion to something else related to microbiome/lung cancer, it makes the article look incomplete. Also, try to restructure the whole manuscript into 3-4 main chapters. Example: Lung microbiome in cancerous and non-cancerous diseases with the the first three subchapters. 

Microbiome as a modulator of cancer therapy response. 

3. Trey to resize and increase the quality of the figures and detail the legents. 

4. Table 1 - mechanism is missing for a big part of microorganisms. Can it be filled from other references? It would be a great assessed to have this table completed. 

5. Please extend the Future direction section by detailing on the current clinical trials available and the potential of reprogramming the microbiome as a new approach of improving the therapy response. 

Minor changes:

1. line 206. abovementioned - above mentioned. 

2. line 250. PDL1 should be PD-L1. 

3. line 325. hosFt - typo.

Author Response

All comments have been addressed. All changes are tracked. We apologize for the delay. It is somehow hard to insert a point by point response at this moment as the manuscript seems entirely new.

Reviewer 2 Report

The Lung Microbiome and Lung Cancer: A Growing Relation ship by Maroun Bou Zerdan, Joseph Kassab, Paul Meouchy, Rami Nehme, Morgan Bou Zerdan, Gracia Fahed and Arun Nagarajan. 

The manuscript describes the pivotal role of microbiota in lung cancer develop, progression and therapy. The paper si well written and organized. The study is interesting and worth to publish. However, there are some issues which have to be addressed. In particular there are some typing errors. E.g. lines 40 and 41. 108 is it probably 108 and cm2 is cm2

Author Response

(The authors gave the same response as above.)

Round 2

Reviewer 1 Report

Dear Authors, 

Thank you for submitting your revised work. I think that you managed to make important improvements since the initial version that are now seen throughout the manuscript. I have to mention that figure 1 still requires further editing as the quality is still low. 

Kind regards,